

# Accounting for forest age in the tile-based dynamic global vegetation model JSBACH4 (4.20p7; git feature/forests) – a land surface model for the ICON-ESM

Julia E. M. S. Nabel[1], Kim Naudts[1,a], and Julia Pongratz[1,b]

[1]Max Planck Institute for Meteorology, 20146 Hamburg, Germany
[a]now at: Department of Earth Sciences, VU University Amsterdam, The Netherlands
[b]now at: Ludwig-Maximilians-Universität München, Munich, Germany

**Correspondence:** J. E. M. S. Nabel (julia.nabel@mpimet.mpg.de, jemsnabel@gmail.com)

**Abstract.** Natural and anthropogenic disturbances, in particular forest management, affect forest age-structures all around the globe. Forest age-structures in turn influence biophysical and biogeochemical interactions of the vegetation with the atmosphere. Yet, many dynamic global vegetation models (DGVMs), including those used as land surface models (LSMs) in Earth system models (ESMs), do not account for subgrid forest age structures, despite being used to investigate land-use effects on the global carbon budget or simulating land–atmosphere interactions. In this paper we present a new scheme to introduce forest age-classes in hierarchical tile-based DGVMs combining benefits of recently applied approaches. Our scheme combines a computationally efficient age-dependent simulation of all relevant processes, such as photosynthesis and respiration, without loosing the information about the exact forest age, which is a prerequisite for the implementation of age-based forest management. This combination is achieved by using the hierarchy to track the area fraction for each age on an aggregated plant functional type level, whilst simulating the relevant processes for a set of age-classes. We describe how we implemented this scheme in JSBACH4, the LSM of the ICON-ESM. Subsequently, we compare simulation output against global observation-based products for gross primary production, leaf area index and above-ground biomass to assess the ability of simulations with and without age-classes to reproduce the annual cycle and large-scale spatial patterns of these variables. The comparisons show differences exponentially decreasing with the number of distinguished age-classes and linearly increasing computation costs. The results demonstrate the benefit of the introduction of age-classes, with the optimal number of age-classes being a compromise between computation costs and accuracy.

## 1 Introduction

Land use, particularly forest management, substantially influences the age structure of global forests (Pan et al., 2011; Erb et al., 2017). More than 19 M km$^2$ of forest area, i.e. about 15% of global ice-free land, are under some kind of management (Luyssaert et al., 2014), with 65% being under regular harvest schemes and another 7% being intensive plantations (Erb et al., 2017). Often, management practices make use of rotation cycles, as common in shifting-cultivation (Boserup, 1966) or even-aged forest management strategies that historically were common in temperate forests and are still the dominant management



type in boreal forests (Kuusela, 1994; Puettmann et al., 2015; Kuuluvainen and Gauthier, 2018). Forest age structures are also influenced by other natural or anthropogenically caused disturbances such as fires, windthrows, droughts, pests and insect outbreaks (e.g. Soja et al., 2006; van Mantgem et al., 2009; Dore et al., 2010; Pan et al., 2011, 2013).

Changes in forest age structure in turn influence biophysical and biogeochemical interactions with the atmosphere, through changes in land surface properties such as albedo, surface roughness, heat fluxes and carbon uptake (e.g. Juang et al., 2007; Dore et al., 2010; Sun et al., 2010; Kirschbaum et al., 2011; Pan et al., 2011, 2013; Poorter et al., 2016; Erb et al., 2017). Forest age structure changes can influence the susceptibility to the environment and to environmental changes. It is, for example, hypothesised that the response of forests to increasing atmospheric $CO_2$ ceases as the forest matures, because other resources than $CO_2$, such as water and nutrients, become growth limiting (Körner, 2006).

It is crucially important to include forest age structures in estimates of the effects of land use on the global scale, primarily due to (1) the aforementioned large extend and substantial effects of forest undergoing changes in age structure; (2) the aim of global studies to include forest management effects in addition to anthropogenic land cover changes; and (3) because global studies usually only have a very coarse resolution. One example for such global studies are the estimates of global land-use emissions for the annual global carbon budgets (Le Quéré et al., 2018), which are conducted with dynamic global vegetation models (DGVMs). Here, 10 out of the 16 participating DGVMs account for wood harvesting. Another example are studies estimating both biophysical and biogeochemical interactions between the land surface and the climate system. These studies are conducted with Earth system models (ESMs) including their land surface models (LSMs), many of which taking part in the coupled model intercomparison projects (CMIPs). Here, considerations of forest age structure might in particular be important for future scenarios that often include strong land-based mitigation measures, such as forest management and afforestation (e.g. in CMIP6's land use intercomparison project LUMIP, see Lawrence et al., 2016). Global studies, in particular the computationally expensive ESM simulations, inevitably need to be conducted on coarse horizontal resolutions, typically only about 0.5° to 2°. Land use will thus usually only happen on fractions of the grid-cells, creating the need to represent subgrid forest age structures. The importance of subgrid forest age structures is also underlined by recent studies stressing the role of forest (re-)growth for the historical and future terrestrial carbon uptake (e.g. Kondo et al., 2018; Krause et al., 2018; Yao et al., 2018) and by studies simulating smaller land use emissions when accounting for secondary forests (e.g. Shevliakova et al., 2009; Yue et al., 2018a). Despite all this evidence, many DGVMs, and particularly also those used as LSMs in ESMs, do not account for subgrid forest age structures (Pongratz et al., 2017).

There are categorically different approaches of how subgrid forest age structures can be implemented in DGVMs, depending on whether these models are individual-/cohort-based or tile-based. In the class of individual- and cohort-based models (referred to as vegetation demographic models in Fisher et al., 2018), subgrid structures are inherently provided. Examples are ED-derivatives (Fisher et al., 2015, 2018), LPJ-Guess (Smith et al., 2001; Bayer et al., 2017), and the SEIB-DGVM (Sato et al., 2007). In the (larger) class of tile-based models (also referred to as area-based in Smith et al., 2001), subgrid structures are not inherently provided. In these models each tile describes an average individual per plant functional type (PFT). Examples for this class of DGVMs are CABLE (Haverd et al., 2018), Class-CTEM (Melton and Arora, 2016), ISAM (Yang et al., 2010), JSBACH (Reick et al., 2013; Mauritsen et al., 2019), LM3 (Shevliakova et al., 2009), LPX-Bern (Stocker et al., 2014b), dif-



ferent versions of ORCHIDEE (Naudts et al., 2015; Yue et al., 2018b) and others. In our study we focus on the second class of DGVMs, as they are more commonly used as LSMs in ESMs. One reason that tile-based models are more commonly used is simply that they have lower computational costs. Another reason is their often historically conditioned top-down development, which facilitates a fully coupled execution within the corresponding ESM.

To extent tile-based DGVMs to represent subgrid forest age structures, two approaches have recently been developed. The most frequently applied approach has been to increase the number of tiles in such a way that a certain number of age-classes or structurally similar stands can be distinguished. A pioneer study has been the paper by Shevliakova et al. (2009), using the model LM3 with a fixed number of in total 12 secondary land tiles for all PFTs and a similarity-based merging of tiles in order to maintain the number of tiles despite further land use/disturbances. Comparably, ORCHIDEE-MICT (Yue et al.,

2018b) introduced a fixed number of six tiles per woody PFT, with tile merging upon exceeding pre-defined woody biomass boundaries. In ORCHIDEE-CAN three tiles per woody PFT with tile merging upon exceeding diameter thresholds have been used, while further within-stand structuring has been applied in each tile by accounting for a user-defined diameter distribution (Naudts et al., 2015). An increase of tiles has also been chosen in ISAM (Yang et al., 2010) and LPX-Bern (Stocker et al., 2014a, b); in these models, however, only one additional tile per PFT has been introduced in order to distinguish primary and

secondary vegetation. A common drawback of the hitherto existing implementations is the missing traceability of the actual age of the forests as soon as tiles are merged. Merging of tiles, however, is a necessity when the number of age-classes is constrained by computational costs.

    The alternative approach for extending the number of tiles to represent subgrid forest age structure in tile-based DGVMs is to keep the information about the forest structure in a separate module. For ORCHIDEE-FM (Bellassen et al., 2010), for

example, ORCHIDEE has been coupled to a forest management module (FFM). FFM takes the tile wood increment calculated in ORCHIDEE as input, allocates the increment to tracked individual trees, conducts self-thinning and forest management, and feeds back the leaf area index (LAI), biomass and litter to the tile. A comparable coupling is described in Haverd et al. (2018), where the DGVM CABLE is coupled to the Population Orders Physiology (POP) module for woody demography and disturbance-mediated landscape heterogeneity (Haverd et al., 2014). POP has a detailed description of the forest structure and

simulates the growth of age/size classes of trees competing for soil resources and light. For each forest tile, POP gets the stem NPP from CABLE and returns woody vegetation height, mortality and sapwood mass. Whilst such a use of a separate module principally enables tracking the exact age of the forest in a grid-cell, it has the drawback of using average tile information to compute land–atmosphere interactions, such as photosynthesis or soil moisture state. Feedbacks between stands of different ages and the environment can thus not be represented.

In this paper we try to bridge the two approaches for extending tile-based DGVMs to represent subgrid forest age in the sense that we present a way to trace the actual age of the forests in a grid-cell despite following the first approach using a restricted number of additional tiles and thus required merges. This approach is therefore more accurate than the current way of implementing subgrid forest age structures by increasing the number of tiles and allows land–atmosphere interactions to be simulated in dependence of forest age. The suggested approach is applicable for any tile-based DGVM, provided the tiles are

structured in a hierarchical way. We describe the implementation of our approach in the DGVM JSBACH4 and use the new



model version to conduct test simulations with different numbers of age-classes and age distribution schemes. Subsequently, we compare the different simulation results against observation-based data to evaluate the compromise between computation costs and accuracy.

## 2 Methods

### 2.1 JSBACH4

The DGVM JSBACH4 is used as LSM in the ICON-ESM (Giorgetta et al., 2018). It is developed with a flexible interface, such that it is also usable within MPI-ESM1.2 (Mauritsen et al., 2019) and as a standalone model driven by climate data. JSBACH4 is a re-implementation of JSBACH3 (Mauritsen et al., 2019) but with a more flexible and extendable structure via a hierarchical representation of tiles (Fig. 1), allowing different processes to be simulated on different levels of the hierarchy. Whilst JSBACH4 is a fully fledged DGVM, and most of the processes from JSBACH3 already have been ported to JSBACH4, some important processes still need to be implemented in the current version (4.20p7), in particular the representation of natural and anthropogenic land cover change (Reick et al., 2013). In order to better be able to represent forest re-growth, the JSBACH4 version (4.20p7) used as basis in this paper has been amended by a dependency of the maximum leaf area index (LAI) on the available leaf carbon, which only recently has been implemented in JSBACH3 (Naudts et al., in prep).

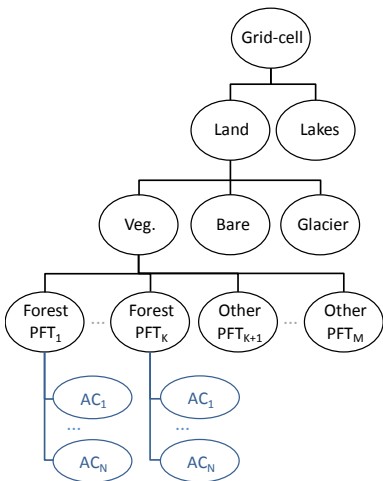

**Figure 1.** The hierarchical tile structure of JSBACH4. In our study, the default tile structure of JSBACH4 (in black) has been extended by a variable number N of forest age-classes (AC) below each of the K forest plant functional types (PFT; in blue).





## 2.2 Subgrid age structure with tracing of forest age – general concept

As outlined in the introduction, we aim for a scheme that allows subgrid forest age structures to be introduced in tile-based models in a computationally efficient way, but with exact tracing of the age of the forest. We thereby follow the mentioned approach of increasing the number of simulated tiles, but taking advantage of a hierarchical organisation of the tiles.

In tile-based models with a flat hierarchy the introduction of a subgrid age structure by increasing the number of tiles would require all processes previously executed on the PFT tile to also separately be simulated on each of the tiles representing age-classes. A hierarchical organisation of the tiles, such as provided in JSBACH4 (Fig. 1), allows for a computationally more efficient way of introducing age-classes, because it makes it possible to choose on which level of the hierarchy processes are simulated. With a hierarchical organisation, different processes and associated state variables can be located on different levels

of the tile hierarchy (Fig. 2). Thus, it is not a prerequisite any more that all processes previously executed on the PFT tile are simulated on each age-class tile. Now, only those processes specific to the development of an age-class, such as photosynthesis and carbon allocation, need to be executed on that age-class. All processes related to several age-classes, such as harvesting, ageing or required merges, are located on the PFT level, which manages associated age-classes. Due to the hierarchy, moreover, meta-information can be maintained on the PFT level. This latter aspect enables to trace the exact age of the forest up to a certain

maximum age $maxAge$ despite only simulating a much smaller number N of age-classes (Fig. 2).

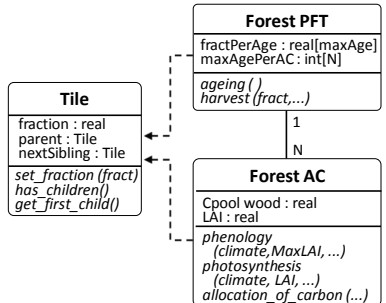

**Figure 2.** UML diagram showing the relation between tiles, forest plant functional types (PFTs) and forest age-classes (ACs). Forest PFTs and ACs are distinct types of tiles in JSBACH4, with each PFT having N associated ACs. Each tile hosts certain state variables, for example the grid-cell fraction that it covers, as well as functions (in italics), for example to navigate the tile hierarchy. Different types of tiles add specific further variables and functions. Tiles representing ACs host variables and functions required for processes calculated on the lowest level of the tile hierarchy, such as photosynthesis or carbon allocation. Tiles representing PFTs host variables to maintain meta-information, for example a vector (fractPerAge) containing the fraction covered by a certain age, i.e. one entry per year up to the maximum tracked age (maxAge). Furthermore, they host functions concerning more than one associated AC, for example forest ageing or harvest. Depicted state variables and processes are exemplary.



## 2.3 JSBACH4–FF

JSBACH4's new git feature "forests" (JSBACH4–FF) introduces an implementation of subgrid age structure following Section 2.2. For this purpose a fixed user-defined number N of age-classes is pre-set in the configuration file for all forest PFTs (Fig. 1). In addition, the upper-bound of each age-class $AC_M$ ($maxA_M$) as well as the total maximum age ($maxAge$) need to

be pre-defined. $maxAge$ determines the oldest age up to which the age per area is tracked, i.e. the length of the $fractPerAge$ vector in each forest PFT (Fig. 2). Area fractions with ages exceeding $maxAge$ are not further distinguished and are refereed to as old-grown forest. For a maximum age of 150 years, for example, each forest PFT would contain a vector with length 150 to track the exact age of the entire forest area up to 150 year old forest. Each $AC_M$ covers a certain interval of years [$maxA_{M-1}$, $maxA_M$) (Fig. 3), with the youngest AC ($AC_1$) always covering the range of year 0 to 1, and the oldest $AC_N$ covering all forest

older than $maxA_{N-1}$, i.e. [$maxA_{N-1}, INF$), with $maxA_{N-1} \leq maxAge$.

In JSBACH4–FF different processes are implemented that can cause shifts of fractions from one AC to another (Fig. 3).

**Ageing** The ageing of forests happens annually and affects the oldest year in each AC, i.e. upon getting one year older the fraction of forests having age $maxA_{M-1}$-1 will shift from $AC_{M-1}$ into $AC_M$ .

**Harvest** In the current version, harvest is implemented as a clear-cut of a fraction of the oldest available AC and can happen annually. Harvest causes a shift of the harvested fraction to the youngest AC.

**Disturbances** The implemented disturbances (wind and fire) can happen daily and are assumed to clear certain area fractions of vegetation, as assumed in JSBACH3 (Brovkin et al., 2009). In JSBACH4–FF disturbances cause shifts of fractions of affected ACs to the youngest AC.

These three processes are managed on the PFT level, where the exact forest age fractions are tracked. Each initiated shift entails a redistribution of forest carbon and a re-determination of other affected state variables of the involved ACs, which are directed and scheduled on the PFT level but exerted on the ACs. When a certain area ($fa$) is moved from one AC to another, e.g. upon ageing, each affected state variable $V_T$ of the target $AC_T$ needs to be updated. $V_T'$ is obtained by weighting the values $V_S$ of the source $AC_S$ and $V_T$ of the target $AC_T$ with respect to the ratio of incoming ($fa$) to current area ($fc$):

$$V_T' = \frac{V_T \cdot fc + V_S \cdot fa}{(fc + fa)} \tag{1}$$

## 2.4 Simulation set-up and measures of model performance

The main purpose of JSBACH are global applications, often in a mode coupled to an ESM. Therefore, we assess the ability of different set-ups without and with different numbers of age-classes to reproduce the annual cycle and large-scale spatial patterns by comparing simulated variables against global observation-based products for different seasons and regions. We

conducted simulations following a protocol described below (Section 2.4.2), with the aim to simulate actual 2010 forest age distributions and forest states. Forest gross primary production (GPP), forest LAI and forest above-ground biomass (AGB)



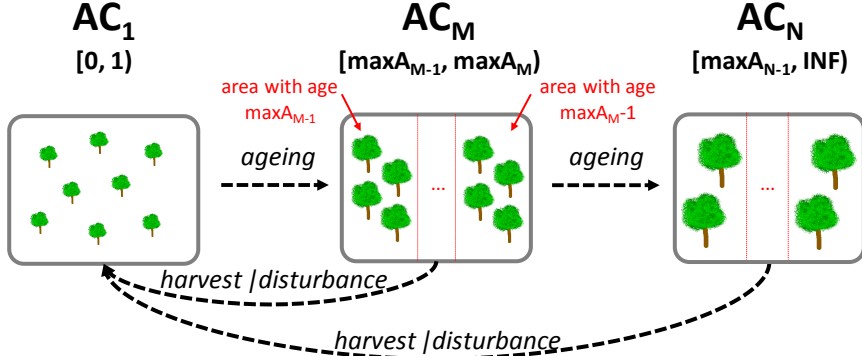

**Figure 3.** Visualisation of forest age-class (AC) boundaries and processes causing shifts from one AC to another in JSBACH–FF. Each AC covers a certain interval of ages. The first AC contains all forest younger than one year; an arbitrary $AC_M$ covers $[maxA_{M-1}, maxA_M)$, i.e. the ages in the right-open interval of the upper boundary of the previous AC ($maxA_{M-1}$) and its own upper boundary ($maxA_M$); finally the last class $AC_N$ covers all forest older than and including the upper boundary of the next younger class ($maxA_{N-1}$), with $maxA_{N-1}$ being smaller or equal to the total maximum age ($maxAge$). The information on the area covered by the different ages (indicated in red) is only known to the associated forest PFT ($fractPerAge$ in Fig. 2). Three processes can lead to movement of area fractions among ACs: ageing leads to movement of the fraction exceeding the maximum age of an AC; harvest and disturbances lead to movement of fractions to the first AC.

simulated for 2001 to 2010 were compared against GPP, LAI and AGB data based on observations using a normalised root mean squared error (NRMSE; Section 2.4.5).

### 2.4.1 Observation-based data

We used GPP and LAI data for the year 2010 as derived in Tramontana et al. (2016). This data already had been mapped to
JSBACH's forests PFTs in a previous study (see supplementary of Nyawira et al., 2016). From this data seasonal means were calculated and expressed per forest area by dividing them by the sum of the forest cover fractions used for the JSBACH4–FF simulations (Section 2.4.2). The AGB per forest area (Avitabile et al., 2016) was downloaded from the GEOCARBON data portal (www.bgc-jena.mpg.de/geodb/projects/Data.phd) and remapped to T63 using the conservative remapping operator of the climate data operators (CDOs, version 1.9.5). Figures S4.2–S4.4 in the supplementary show maps of the pre-processed
observation-based data.

### 2.4.2 General simulation set-up

We conducted simulations with JSBACH4 (4.20p7) feature/forest in standalone mode hosted within the MPI-ESM environment (see supplementary material S1 for further information). We used JSBACH's default set-up comprising 12 PFTs, of which 4 are of a forest type: Tropical evergreen and deciduous forest (TE and TD) and extratropical evergreen and deciduous forest (ETE
and ETD). Simulations started in 1860 from scratch, i.e. with empty vegetation carbon stocks, and were run up to 2010. We





used T63 resolution (192 longitudes x 96 latitudes; $1.9° \times 1.9°$), a climate forcing based on GSWP3 (Kim et al., 2012) and $CO_2$ from the collection of greenhouse gas concentrations for CMIP6 (Meinshausen et al., 2017). To obtain forest age distributions comparable to those observed for 2010 (Poulter et al., 2018), harvest was conducted following prescribed maps (Section 2.4.3) and natural disturbances were switched off in order to not additionally alter forest age. The simulations were conducted with

a static land-use map for 2010, based on TRENDYv4 JSBACH3 output (Le Quéré et al., 2015). The TRENDYv4 JSBACH3 simulation started from a potential vegetation map extrapolated from remote sensing (Pongratz et al., 2008) and was forced by the Land-use Harmonization dataset LUH1 (Hurtt et al., 2011). We conducted simulations with different numbers and distributions of age-classes (Section 2.4.4). All simulations were conducted on Mistral, the High Performance Computing system of the German Climate Computing Center (DKRZ), using an identical number of CPUs.

### 2.4.3 Harvest maps

Harvest maps were derived such that the observed 2010 forest age distribution given by Poulter et al. (2018) is reached in the final simulation year 2010. The observed forest age map of course not only reflects forest harvest, but all processes influencing the age of a forest, i.e. also natural disturbances and anthropogenic land cover change. Because assigning the observed age structure to forest harvesting vs natural disturbances vs anthropogenic land cover change would come with uncertainties and

is not relevant for our study (as only the affected fraction of an age-class matters, independent of the underlying causes), we apply only forest harvest to obtain the observed age distribution.

The map by Poulter et al. (2018) provides the global forest age distribution of 4 forest PFTs (needleleaf evergreen and deciduous, as well as broadleaf evergreen and deciduous) on a grid with 0.5°resolution. The map uses a discretisation into 15 age-classes, covering 10 years each, with the last class containing all area with an age >140 years. In a pre-processing step,

the map was remapped to T63 using the conservative remapping operator of the CDOs. Subsequently, the PFTs from the map were scaled to JSBACH's PFT cover fractions. From these scaled maps, we derived harvest maps for each simulation year such that the simulated age distribution conforms with the observed one in 2010, assuming that the fractions given by Poulter et al. (2018) are equally distributed over the ten years covered by each age-class (see supplementary material S2 for more details). In each simulation year the to be harvested fraction was read from the harvest map for that simulation year and the according

fraction was transferred from the oldest to the youngest forest age. In the first (1860) and in the last simulation year (2010) no harvest was conducted.

### 2.4.4 Simulated number of age-classes and age distribution schemes

Table 1 lists the conducted simulations. We used different numbers of age-classes and two different age distribution schemes described below. The selected numbers of age-classes are exemplary only; however, the finest resolution into 15+1 age-classes

was motivated by the discretisation of the age map (Section 2.4.3). In addition to simulations with age-classes, we performed one simulation only using PFTs, i.e. without age-classes. In this simulation we used the same harvest fractions prescribed as in the simulations with age information, but harvest was applied as done in JSBACH3 (Reick et al., 2013), i.e. by simply diluting the wood carbon of the harvested PFT tiles.





The two applied age distribution schemes are defined as follows:

**EAS** The equal age-spacing (EAS) distribution scheme spreads the age-classes evenly over the age space. For example, a maximum traced age of 150 years distributed evenly over 10+1 age-classes (EAS11 in Table 1) results in age-classes covering 15 years with the following upper age bounds: 1, 16, 31, 46, ..., 136, INF. This distribution was motivated by the equal spacing applied in the forest age map by Poulter et al. (2018).

**IAS** The increasing age-spacing (IAS) distribution scheme uses an increasing age range covered with increasing age, i.e. younger age-classes cover smaller intervals in the age space than older age-classes. The upper age bound of a forest age-class M ($uLim_M$) is defined following Eq. 3.

$$spacing = \frac{maxAge}{\sum_i^{N-1} i} \tag{2}$$

$$uLim_M = \begin{cases} 1, & \text{if M=1} \\ INF, & M = N \\ uLim_{M-1} + int(spacing \cdot M), & \text{else} \end{cases} \tag{3}$$

With $maxAge$ being the maximum age and N being the number of age-classes. A maximum age of 150 years distributed with IAS over 10+1 age-classes (IAS11 in Table 1) results in age-classes with the following upper bounds: 1, 3, 8, 16, 26, 39, 55, 74, 95, 119, INF. This second distribution scheme was motivated by the fact that young forests usually have larger incremental changes in most variables than old ones (see e.g. Amiro et al., 2006; Martínez-Vilalta et al., 2007; Leslie et al., 2018).

Both distribution schemes are applied in a static way, i.e. the age-class boundaries do not change during runtime. Figure 4 shows the division into age-classes for the different simulation set-ups for an example grid-cell in Canada.

**Table 1.** Conducted simulations with number of age-classes and applied age distribution scheme. The "+1" in the number of age-classes refers to the youngest age-class, which always covers the years 0 to 1 in JSBACH4–FF.

| Simulation name | PFT | EAS03 | IAS03 | EAS06 | IAS06 | EAS08 | IAS08 | EAS11 | IAS11 | EAS13 | IAS13 | EAS16 | IAS16 |
|---|---|---|---|---|---|---|---|---|---|---|---|---|---|
| Number of age-classes | – | 2 + 1 | 2 + 1 | 5 + 1 | 5 + 1 | 7 + 1 | 7 + 1 | 10 + 1 | 10 + 1 | 12 + 1 | 12 + 1 | 15 + 1 | 15 + 1 |
| Age distribution scheme | – | EAS[a] | IAS[b] | EAS | IAS | EAS | IAS | EAS | IAS | EAS | IAS | EAS | IAS |

[a] EAS: equal age-spacing; [b] IAS: increasing age-spacing

### 2.4.5 NRMSE

We calculated the area-weighted root mean squared error (RMSE) according to Eq. 4 based on difference maps between 'OBS', the observation-based data (Section 2.4.1), and 'SIM', the results of each simulation (see Table 1). The RMSE was calculated for 2001-2010 simulation output averages, separately for each variable 'V' and three selected regions 'R'. Each selected region



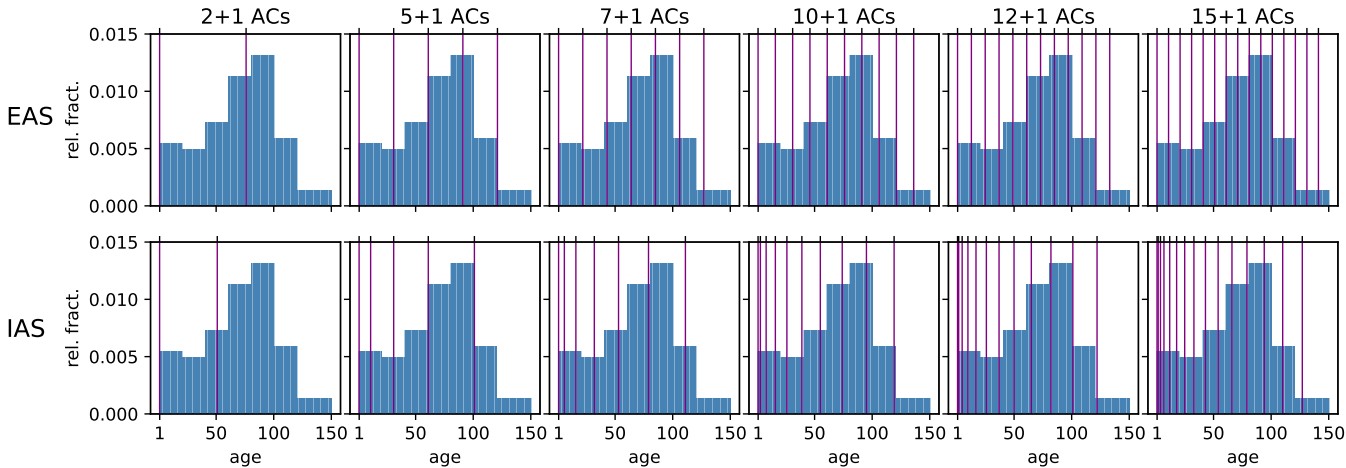

**Figure 4.** Division into age-classes (ACs) for the different simulations listed in Table 1 (EAS: equal age-spacing; IAS: increasing age-spacing). Purple lines mark the upper age boundary of each age-class. The blue bars show the relative fraction for each year resulting for an example cell in Canada (lon = 286.875, lat = 47.5639) in the simulation year 2010. Note that no harvest was conducted in the final simulation year 2010, therefore the smallest age-class is empty.

defines a latitudinal band, including all forested land on a subset of the 96 latitudes and the entire 192 longitudes (see Table 2 for the regions, their latitudinal boundaries and the latitude indices b1 and b2). For GPP and LAI the four seasons 'S' (DJF, MAM, JJA, SON) were calculated separately. The RMSE for each variable, region and season was subsequently normalised with the range (Max-Min) observed for that variable, region and season (Eq. 5).

**Table 2.** Selected regions used for the comparison of simulation results and observation-based data and their latitudinal boundaries and indices.

| Abbr. | Region | max lat | b1 | min lat | b2 |
|-------|--------|---------|-----|---------|-----|
| BOR | Boreal | 90° | 1 | 50° | 21 |
| NH-TMP | Northern Hemisphere Temperate | 50° | 22 | 30° | 32 |
| TROP | Tropical | 30° | 33 | -30° | 64 |

$$5 \quad \text{RMSE}_{V,S,R} = \sqrt{\frac{\sum_{k=1}^{192} \sum_{m=b1}^{b2} \left( (\text{OBS}_{V,S,\text{lon}(k),\text{lat}(m)} - \text{SIM}_{V,S,\text{lon}(k),\text{lat}(m)})^2 \cdot \frac{\text{AREA}_{\text{lon}(k),\text{lat}(m)}}{\text{AREA}_R} \right)}{192 \cdot (b2\text{-}b1+1)}} \quad (4)$$

$$\text{NRMSE}_{\text{Max-Min},V,S,R} = \frac{\text{RMSE}_{V,S,R}}{\text{Max}(\text{OBS}_{V,S,R}) - \text{Min}(\text{OBS}_{V,S,R})} \quad (5)$$





To more easily assess changes in performance when increasing the number of age-classes the different NRMSE$_{\text{Max-Min}}$ values were subsequently aggregated per variable by averaging over the regions (for AGB) and in addition over the seasons (for GPP and LAI), using equal weights.

### 2.4.6  Computational costs

In addition to determining the NRMSE for different variables, we also determined the computation costs of the different set-ups. We calculated the average CPU time recorded for the simulation years 2001-2010. Whilst absolute computation times are of less interest here, particularly since JSBACH4 is still highly under development and currently does not reach the targeted performance, relative differences among the set-ups depict the costs of the introduction of subgrid forest age structures.

## 3  Results and Discussions

Having forest age-classes in JSBACH4–FF facilitates a finer discretisation in each grid-cell and enables the implementation of age-based forest management. The number of age-classes in JSBACH4–FF is flexible, and in the following we describe the evaluation of simulation results using different numbers of age-classes and age distribution schemes and discuss the compromise between computation costs and accuracy, when selecting a certain number of age-classes (Section 3.1). Subsequently, we more closely examine differences between a simulation with an exemplary number of age-classes and a simulation only

using PFTs, i.e. without age-classes, to investigate the benefits of having age-classes in JSBACH4–FF (Section 3.2). Finally, we discuss assets and drawbacks of alternative schemes introducing age-classes in tile-based DGVMs (Section 3.3).

### 3.1  Evaluation

In this section we use the NRMSE$_{\text{Max-Min}}$ for different regions/seasons as aggregated measure to compare the different simulation set-ups. A closer examination between a simulation with and without age-classes including a spatially explicit comparison

follows in Section 3.2.

Introducing age-classes improves the comparison to observation-based data for nearly all compared variables, regions and seasons (Fig. 5), with the only notable exception of the AGB in the boreal region, where the PFT simulation was more similar to the observation-based data than the simulations with age-classes (Fig. 5c). For most comparisons, the NRMSE$_{\text{Max-Min}}$ indicates a small but distinct improvement over not representing a forest age structure for all simulated numbers of age-classes and both

age distribution schemes.

Averaging the NRMSE$_{\text{Max-Min}}$, giving each region and each season the same weight, results in an NRMSE$_{\text{Max-Min}}$ exponentially decreasing with the number of age-classes for GPP and LAI (Fig. 6a and b). This shape holds for all regions, with a faster decrease and an earlier saturation for the northern hemisphere temperate and tropical regions than for the boreal region (Fig. S3.1a-f). The NRMSE$_{\text{Max-Min}}$ for AGB shows a slowly saturating increase with the number of age-classes for the bo-

real region (Fig. S3.1g) and only small differences among the different numbers of age-classes in the northern hemisphere temperate and the tropical regions (Fig. S3.1h and i). The observed increase in NRMSE$_{\text{Max-Min}}$ for the boreal AGB is due to





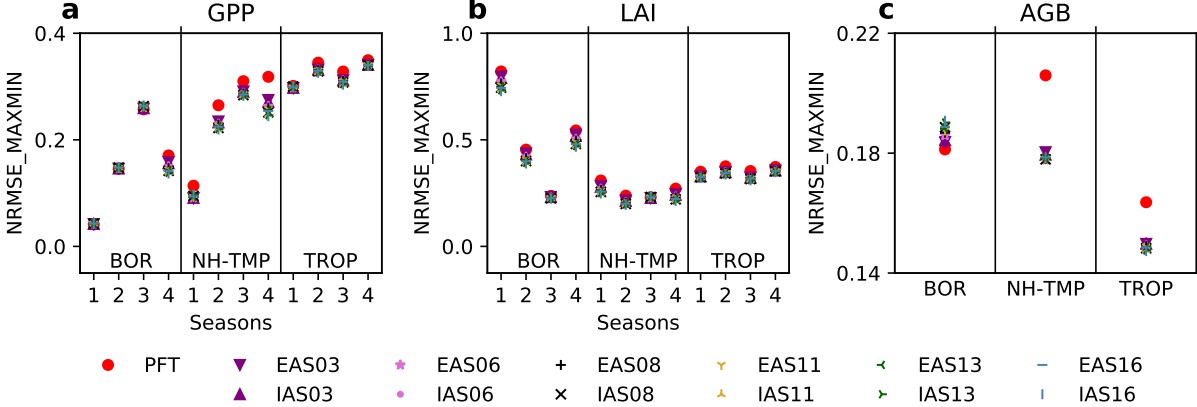

**Figure 5.** Evaluation of the conducted simulations (Table 1) with observation-based data by means of the $NRMSE_{Max-Min}$ (Section 2.4.5). Depicted are calculated $NRMSE_{Max-Min}$ (Section 2.4.5) values for each simulation for the gross primary production (GPP; panel **a**), the leaf area index (LAI; panel **b**) and the above-ground biomass (AGB; panel **c**). The $NRMSE_{Max-Min}$ is calculated as the root mean squared error of observation-based data and simulation results, normalised with the range (Max-Min) of the according variable for each of the selected regions (Table 2); and for LAI and GPP also for each of the four seasons.

an increased underestimation when accounting for more young forest, as also discussed below (Section 3.2). Apart from the boreal AGB comparison, all comparisons show a smaller $NRMSE_{Max-Min}$ for simulations using the IAS distribution scheme (Fig. 6 and Fig. S3.1), i.e. a distribution applying an increasing age space (visualised in Fig. 4). This increase in performance is due to the finer discretisation of younger age-classes with fast-changing LAI and GPP, which saturate for older age-classes (see

e.g. Fig. 7 for GPP). In summary, a finer discretisation, particularly of the younger age-classes, is leading to values closer to the observation-based data, albeit the benefit of increasing the number of age-classes is slowly saturating towards larger numbers of age-classes (Fig. 6).

We performed the averaging of the $NRMSE_{Max-Min}$ to more easily assess the differences in performance among the different numbers of age-classes and the two age distribution schemes. For this, we equally weighted the selected regions, because

we wanted to equally account for these regions, which strongly differ in simulated PFTs and land–atmosphere interactions. Alternatively, we could have weighted the regions by area, which would have lead to an increasing weight of the tropical region, and thus to an earlier saturation of the $NRMSE_{Max-Min}$ with increasing age-classes.

Comparisons of required CPU times show a linear increase with an increased number of age-classes (Fig. 6d) and neither a difference between the two age distribution schemes, nor an offset as compared to the PFT simulation. This behaviour

was expected, since the processes requiring most of the computing time, such as the calculation of photosynthesis, carbon allocation and respiration, are conducted on the age-classes. The absence of an offset comparing the PFT simulation with the age-class simulations indicates that the introduced organisational overhead on the PFT level in simulations with age-classes is not substantial.



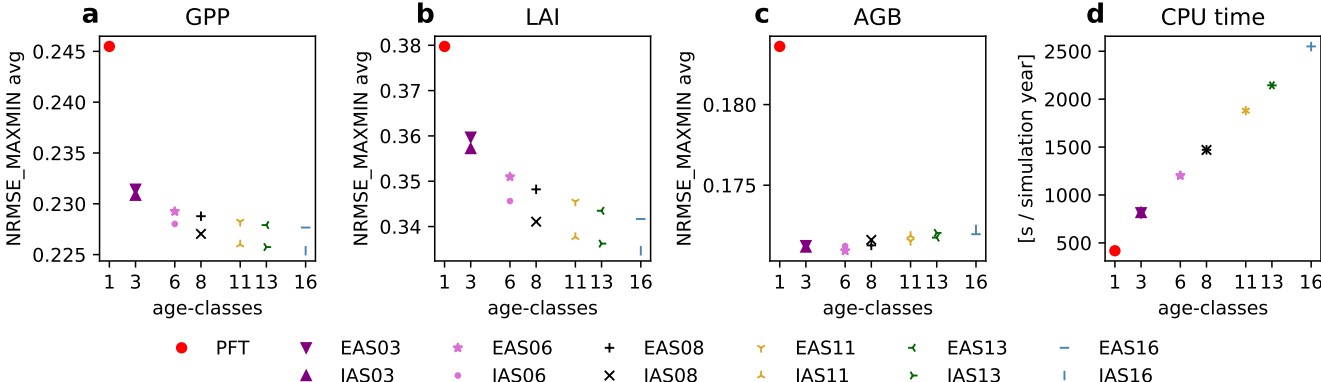

**Figure 6.** Change in NRMSE$_{\text{Max-Min}}$ (Section 2.4.5) and CPU time when increasing the number of age-classes. Panel **a** to **c** show the averaged NRMSE$_{\text{Max-Min}}$ (see also Fig. 5). Averaging has been conducted giving equal weights to all selected regions (Section 2.4.5) for AGB (panel **c**) and in addition to all four seasons for GPP and LAI (panel **a** and **b**). Figure S3.1 in the supplementary material shows the same data separately for each region. Panel **d** shows the computation time required per simulation year averaged over the years 2001-2010.

As expected, the optimal number of age-classes is a compromise between computation costs and accuracy, which is a logical and commonly observed aspect when dealing with discretisation in models (see e.g. Nabel, 2015; Fisher et al., 2018). In the end, the choice of the number of age-classes to be used in a JSBACH4–FF simulation will depend on the application. Simulations comparing different forest management regimes in detail might, for example, aim for a fine discretisation, while more general

5  simulations covering long time-spans might tend to aim for fewer age-classes. For the remaining parts of this manuscript, one set-up has been selected as an illustrative example: IAS11 (see Table 1), i.e. the simulation with 10+1 age-classes and the age distribution scheme with increasing age space. This set-up is a compromise between accuracy of GPP and LAI representation on one hand and boreal AGB representation and CPU time on the other. However, the main findings will not depend on the exact number of age-classes selected, particularly not as long as they are in the saturation part of the exponential decreasing

10  function regarding GPP and LAI comparisons.

### 3.2  On the benefit of having age-classes

The evaluation with observation-based GPP, LAI and AGB data showed that simulations with age-classes were closer to the observation-based data for nearly all comparisons (Fig. 5 and Fig. 6). Spatially explicit comparisons of the results from the PFT simulation and observation-based data ("OBS-PFT" in Figs. S4.2–S4.4, column 2) indicate several areas of underestimation (red) and of overestimation (blue) for all variables. In Fig. 7 we compare results of a JSBACH4–FF simulation with age-classes, a simulation only using PFTs, as representative for a DGVM without forest age structures, and the observation-based data of summer (JJA) GPP. The comparison is done for illustrative grid-points that were selected to cover areas of both over- and underestimation and to represent different typical land-use histories or forest management regimes, resulting in different age distributions: a grid-point with an age distribution matching historically continuous clear-cuts and some more recent changes





in land-use intensity in Germany (Fig. 7a); a grid-point with uniform age distribution resulting from a continuous, steady clear-cutting in Finland (Fig. 7b); a grid-point with untouched old-grown forest on one hand and young managed forest on the other hand in India (Fig. 7c); a grid-point with intensive harvest/disturbances in the south-east of the US (Fig. 7d); a heavily deforested example in east South America resulting nearly exclusively in young forest (Fig. 7e); a grid-cell with

recent afforestation in China (Fig. 7f) and a grid-cell with pre-dominantly old-grown forests in central Africa. In general, the simulation with age-classes results in smaller GPP, LAI and AGB values (Fig. 7 and Figs. S4.2–S4.4, column 3), which is expected, since GPP, LAI and AGB are non-linearly increasing and saturating with age (see e.g. Fig. 7). Therefore, a harvested age-less forest in the PFT simulation has higher values for these variables than a fraction weighted average of an age-structured forest in the same grid-cell in the simulation with age-classes (Fig. 7). Since the simulation with age-classes generally results

in smaller GPP, LAI and AGB values, overestimations can get alleviated, causing a decrease in the $NRMSE_{Max-Min}$, while underestimations can get more severe, causing an increase in the $NRMSE_{Max-Min}$. The comparison of the differences between observation-based data (OBS) and the PFT simulation results on one hand, and OBS and the IAS11 simulation results on the other hand, accordingly shows higher similarity in several areas where the PFT simulation indicated overestimation (areas which are blue in column 2 and 4 in Figs. S4.2–S4.4) and less similarity in some areas with underestimation (areas which are

red in column 2 and 4 in Figs. S4.2–S4.4). Fig. 7 shows several grid-point examples with increased underestimations of summer GPP (Fig. 7 panel b and c), reduced overestimations (panel a and d) and grid-points where the previous overestimation is now replaced by a slight (Fig. 7e) or an equally large underestimation (Fig. 7f). Globally, reduced overestimations get particularly visible for LAI in the east of South America, and for several seasons also for example over China, North America and Europe (Fig. S4.3d,h,i,p). For GPP (Fig. S4.2d,h,i,p) and AGB (Fig. S4.d) the pattern is more mixed, with reduced overestimations

particularly in the east of North America and China and partly for the east of South America. In addition, there are several areas of under- and overestimation which are very similar in the two simulations (areas coloured in column 2 and white in column 4 in Figs. S4.2–S4.4). These are particularly areas with pre-dominantly old-grown forests, i.e. without a distinct age-structure, such as central Africa, central Amazon and Siberia, where the PFT and the age-class simulation led to similar results (see e.g. Fig. 7g).

Importantly, besides the slight increase in accuracy, the main gain of JSBACH4–FF is the additional functionality by the newly implemented forest age-structure. The age-structure facilitates keeping the coarse resolution required in ESM simulations while nevertheless capturing some of the sub-grid scale heterogeneity that is important to better resolve several of the simulated processes. Particularly, the forest age-structure enables the implementation of different forest management regimes while simultaneously accounting for differences in the productivity and the standing stocks. The grid-point examples shown in

Fig. 7 highlight the relevance of a distinction of age-classes, since they demonstrate the non-linear relationship between GPP and forest age. A similar relationship can be found for AGB and LAI. Consequently, the ability to distinguish age-classes enables a more accurate simulation of the biogeochemical consequences of land use and particularly prescribed harvest regimes. For example, harvesting of younger age-classes will lead to lower land-use emissions, as also described in other studies (e.g. Shevliakova et al., 2009; Yue et al., 2018a). Similarly, being able to distinguish forest age-classes will also affect biophysical



land–atmosphere interactions, since younger forests, for example, have lower LAI and higher albedo (e.g. Bright et al., 2013). A constantly thinned age-less forest will therefore always lead to a lower albedo than a young forest regrowing after a clear-cut.

### 3.3 Limitations and alternative schemes

In the previous sections we compared a JSBACH4–FF simulation when only using PFTs with simulations including forest age-
classes and discussed associated trade-offs and benefits. In this section we discuss limitations and advantages of the applied and of alternative schemes.

Since JSBACH is a tile-based DGVM, the introduction of an individual/cohort-based approach as used in some other DGVMs (e.g. Sato et al., 2007; Fisher et al., 2015; Bayer et al., 2017) would be very complex. Regarding forest age-structures these models have the essential advantage of naturally providing forest demography (Fisher et al., 2018). Due to their com-
plexity, however, they are less commonly used as fully coupled LSMs for ESMs. Being fully coupled with an ESM, however, is one major aspect and purpose of JSBACH, which historically has been part of the MPI-ESM (Mauritsen et al., 2019) and now also is part of the ICON-ESM (Giorgetta et al., 2018).

For tile-based DGVMs, there is at least one option mentioned in the literature that provides an alternative to simply increasing the number of tiles: the coupling of a separated module dealing with the woody demography (see e.g. Bellassen et al., 2010;
Haverd et al., 2018). On one hand, this approach shares the advantage with individual/cohort-based DGVMs that it provides a forest demography and thus principally enables the tracking of forest age. On the other hand, this approach has the important limitation of still calculating key processes concerning the land–atmosphere coupling at the aggregated tile level, i.e. in this approach, processes such as photosynthesis and respiration, are not computed for separate age-classes. This restriction impairs the calculation of biogeochemical and biophysical interactions, due to the non-linearity of forest growth and the associated
non-linear relationships of those key processes with forest age (as e.g. depicted for JJA GPP in Fig. 7). This limitation can only be avoided by increasing the number of tiles.

Building on the approach of increasing the number of tiles, the scheme suggested in this paper adds one important benefit of the alternative schemes by explicitly tracking forest age. It thereby enables the implementation of age-based forest management schemes that historically were common in temperate forests and are still the dominant management type in boreal forests
(Kuusela, 1994; Pan et al., 2011; Puettmann et al., 2015; Kuuluvainen and Gauthier, 2018). Another advantage of the explicit tracking concerns the discretisation error. While the presented approach does require frequent area-weighted merges in order to maintain a limited number of age-classes, it only requires to shift the actually affected parts of an age-class, and not entire age-classes/"cohorts" as common in previous applications (e.g. Shevliakova et al., 2009; Yue et al., 2018b). Upon ageing, for example, in our approach only those fractions of an age-class will be shifted that are actually at the age-limits of an age-class.
With regard to previous studies that increased the number of tiles in order to introduce a more detailed representation of the forest state, our evaluation indicates that the number of additional tiles used in previous applications might have been too coarse. Solely separating primary and secondary forests (e.g. Yang et al., 2010; Stocker et al., 2014a, b) or introducing only a few age-classes/"cohorts" (e.g. Shevliakova et al., 2009; Yue et al., 2018b) might not be sufficient to discretise non-linear





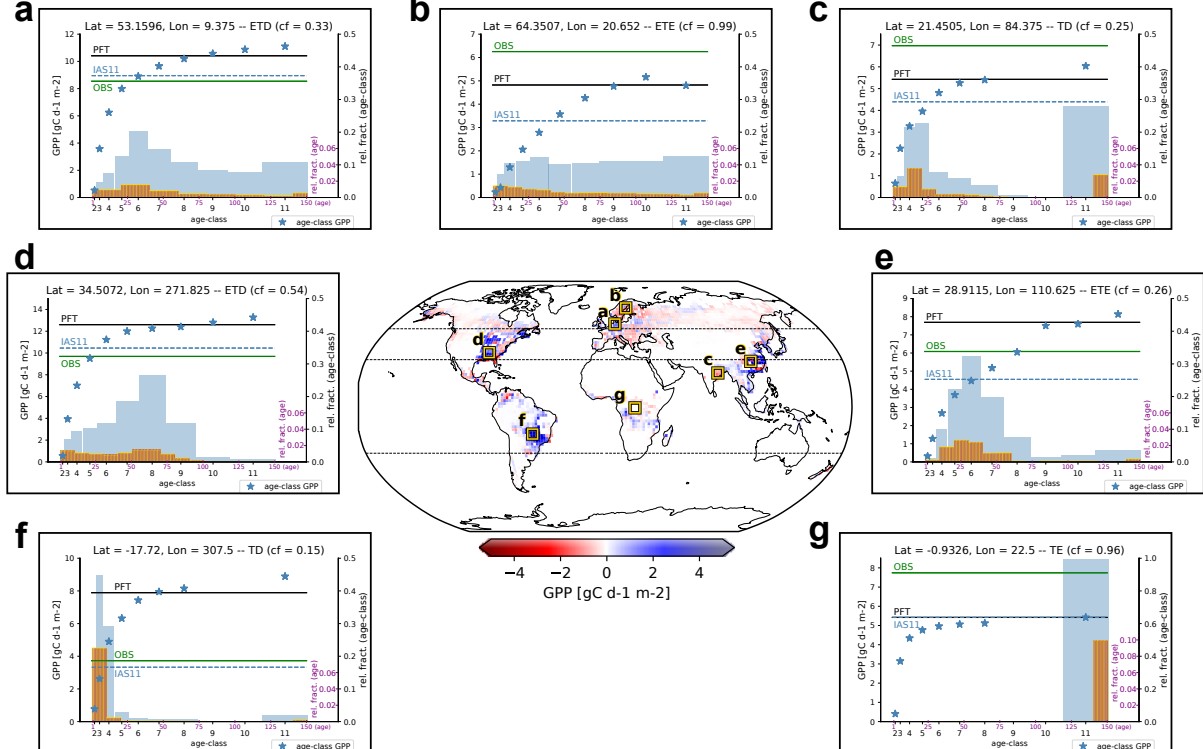

**Figure 7.** Exemplary grid-points comparing results of a simulation using age-classes to those of a PFT simulation and to the observation-based data. Shown are 2001-2010 mean summer gross primary production (JJA GPP) from observation-based data compared to the JJA GPP resulting in two different simulations (PFT and IAS11). The map in the center shows the difference between the absolute differences of the observation-based JJA GPP per forested area and those of the simulations (abs(OBS-PFT)-abs(OBS-IAS11)), i.e. it shows where the results from the simulation with age-classes (IAS11) deviate less (blue) or more (red) from the observation-based data than the PFT simulation results (see also Figs. S4.2-S4.4, column 4). Dashed lines in the map mark the three selected regions – boreal, northern hemisphere temperate, and tropical (see Table 2). The plots (a-g) show the JJA GPP per PFT area (ETD: extratropical deciduous; ETE: extratropical evergreen; TD: tropical deciduous; TE: tropical evergreen) and the area fractions per age-class and per year at the different grid-points framed on the map. For each grid-point the center latitude and longitude, as well as the grid-cell cover fraction (cf) of the depicted PFT are specified. The x-axis reflects the age from 0-151 (purple) with the age-classes (black) indicated at the age centres. The left y-axis depicts the amount of JJA GPP. Stars mark the JJA GPP per age-class, the dashed blue line the resulting JJA GPP value of the plotted PFT in the IAS11 simulation. The black line marks the JJA GPP value resulting from the PFT simulation, i.e. the simulation without age-classes. The green line marks the 2010 JJA GPP value from the observation-based data. The right y-axes depict the 2010 area fractions per age-class (black) and per year (purple) relative to the plotted PFT. Blue bars depict the fraction of each age-class (i.e. one separate bar per age-class) and the yellow framed purple bars the fraction of each age (i.e. one separate bar per year). Note: 1. The age-class JJA GPP is only depicted for age-classes having non-zero fractional cover over the whole timespan 2001-2010 (this is not the case for the age-classes 9 and 10 in panel c,f and g). 2. Age and age-class fractions of classes 2-8 in panel g are very small and therefore not visible above the x-axis. 3. Since we did not apply any harvest in the final simulation year 2010, the first year and accordingly the youngest age-class are always empty.





relationships with forest age (see e.g. Fig. 7, and also Fig. 6), at least not on the coarse resolutions that are common in global model studies dealing with human land use (e.g. Le Quéré et al., 2018).

In this paper, we presented two different approaches to distribute the age space onto the available age-classes: the equal age distribution scheme EAS, which spreads the age-classes evenly, and IAS, a scheme that increases the age space with

increasing age. The evaluation indicated the second approach to be superior to the first (Fig. 6), which can be explained by the finer discretisation of younger age-classes that more accurately resolves the steep part of the non-linear age-dependent relationship of GPP, LAI and AGB (see e.g. Fig. 7). There are, however, other possible age distribution schemes. One could, for example, use smaller age-classes for old ages in addition to the smaller age-classes used for young ages in the IAS scheme. With such a scheme, one could better cover age-related declines as indicated in Fig. 7b or described in Zaehle et al. (2006) and

Bellassen et al. (2010). Another possibility would be to replace the static distribution schemes that are equally applied to all grid-cells with a dynamic scheme creating individual distributions for each grid-cell. In such a dynamic scheme, age-classes could be defined depending on the demand for each grid-cell, with merging based on similarity criteria (see e.g. Shevliakova et al., 2009), i.e. those age-classes would be merged that share the most similar values for a selected variable (e.g. GPP). Such an approach could potentially decrease the discretisation error, particularly for cells with only infrequent disturbances/harvest

events. A drawback could be an increase in the organisational overhead caused by the similarity tests required for each merging step. However, the additional computational effort is not expected to be very large, considering that for now the organisational overhead seems to be very small (linear increase without an offset as shown in Fig. 6d) and particularly since in the current set-up dynamic merges would only be required once a year.

## 4   Summary and Outlook

In this paper we described a new scheme to introduce forest age-structure in a hierarchical tile-based DGVM and presented its implementation in JSBACH4. JSBACH4–FF allows land–atmosphere interactions to be simulated in dependence of forest age and, simultaneously, to trace the exact forest age, enabling the implementation of age-based forest management schemes in JSBACH4–FF.

JSBACH4 itself is still highly under development regarding infrastructure and processes integrated from JSBACH3. In the

version used for this paper (4.20p7), particularly the representation of natural and anthropogenic land cover change has not yet been ported from JSBACH3. Upon implementation, new processes will have to be integrated with the age structure. In addition, other developments would be desirable: harvest, for example, has so far only been implemented as area clear cuts, following the implementation of other disturbances in JSBACH3 (see Brovkin et al., 2009). For a representation of different forest management strategies including intermediate thinning before a final felling, an implementation of forest thinning would

be required (Otto et al., 2014; Naudts et al., 2015). Anthropogenic thinning could be implemented in JSBACH4–FF by keeping the number of individuals as a state variable for each age-class that is manipulated upon thinning, with anthropogenic thinning overruling the already implemented self-thinning.





Despite planned and potential extensions, together with the newly implemented age-classes, JSBACH4–FF already now provides a valuable tool to study forest management effects, particularly due to its integration with the ICON-ESM.

*Code and data availability.* The hosting MPI-ESM model version (MPI-ESM 1.2.01p1) is made available under a version of the MPI-M Software License Agreement and can be obtained after registration from https://www.mpimet.mpg.de/en/science/models/mpi-esm/users-forum/.

5    Data and scripts used in the analysis, the JSBACH4 (4.20p7; git feature/forests) code, a patch to the hosting MPI-ESM required to run JSBACH4–FF as well as other supplementary informations are archived by the Max Planck Institute for Meteorology (https://pure.mpg.de/pubman/faces/ViewItemFullPage.jsp?itemId=item_3032727) and can be obtained by contacting publications@mpimet.mpg.de.

*Author contributions.* JN, KN and JP initiated the implementation of age-classes in JSBACH. JN planned and conducted the implementation, designed and performed the simulations and wrote the first draft of the paper. All co-authors contributed to the analysis and edited the

10    manuscript.

*Acknowledgements.* This work has been supported by the German Research Foundation's Emmy Noether Program (PO1751/1-1). All simulations have been conducted at the German Climate Computing Center (DKRZ; allocation bm0891). The authors like to thank Stiig Wilkenskjeld for reviewing the manuscript prior submission, Veronika Gayler for support regarding JSBACH3, Reiner Schnur for support regarding JSBACH4, and Thomas Raddatz for discussions regarding the implementation of the age-classes.



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
