# Peer review of "Accounting for forest age in the tile-based dynamic global vegetation model JSBACH4 (4.20p7; git feature/forests) – a land surface model for the ICON-ESM"

_Geoscientific Model Development, 2019_

## Referee Comment (RC1) · Anonymous Referee #1 · 8 May 2019

This manuscript presents an interesting and likely significant update to the dynamic global vegetation model JSBACH4. Specifically, a methodology is presented for more accurately simulating forest age structure. It is found that the new model version mostly has lower predictions of gross primary production, aboveground biomass, and leaf area index than the old model version. These lower predictions typically bring the model into closer agreement with the observations. Given the global importance of land-use change and forest harvesting, this paper presents an important step forward. However, there are several areas where I think the manuscript can be improved.

[Figure]

Specific issues

1. The text raises two important expectations that were not fulfilled in the results. First is the issue of land-atmosphere interactions. I would argue that it is conventional to think of land-atmosphere interactions in terms of energy and water budgets, and indeed some authors (e.g., Santanello Jr et al. 2018) define land-atmosphere interactions exclusively in terms of energy and water budgets. Yet the current manuscript contains no results related to energy and water budgets. Personally, I think that such results would be an interesting addition to the paper. Without such an addition, I think the authors need to change the text to more specifically refer to carbon fluxes and budgets. Second is the issue of forest management. The authors repeatedly state (see P11, L10-11 for one example) that an advantage of the model is in simulating different forest management scenarios. However, this is not exploited in the paper (I know that the forest management scenarios in Fig. 7 are different, but so is the climate, so one cannot isolate the effect of the management scenario). Why not illustrate the power of the new modelling approach by running different forest management scenarios for a single grid cell?

2. Some of the methods were not adequately justified. First, I am concerned that the initial condition is unrealistic. Why did simulations begin in 1860 from bare ground rather than from a spin-up? Of course, the 1860 initial condition would more realistically be represented by many forested areas. Second, I could not determine from the paper how one goes from the 2010 age-class distribution to time-dependent (1860-2010) harvest rates. This procedure should be described. Third, it seems like there is an inconsistency between the definition of the model's PFTs (tropical evergreen and deciduous, extratropical evergreen and deciduous) and the Poulter et al. PFTs (broadleaf evergreen and deciduous, needleleaf evergreen and deciduous). What is the correspondence?

3. The comparison to observations can be made more substantial. RMSE is a helpful statistic, but I wonder what is being missed by only considering this statistic. For example, I wonder what can be learned from Taylor diagrams? I am certainly not asking that the paper include Taylor diagrams for every variable, but rather such diagrams could be analyzed in a preliminary analysis and the most exciting ones presented in the paper or supporting information.

4. There are some problems with the interpretation of the results. First, I think it misses the point to repeatedly state that the new model is better. Rather, the fundamental result is that new model tends to reduce GPP and LAI relative to the old model. The new model is better because the old model was biased high. If the old model had been unbiased, then the new model would have been biased low. Alternatively, suppose that there is another modeling group excited by this study, and that that modeling group has a model that is biased low. Then implementation of this scheme would probably make that model worse. Second, I think that more care needs to be taken in the interpretation of Figure 6. While the curves in panels a-c are decreasing, the authors do not quantitatively support their assertion that the curves are decreasing exponentially (and not, say, quadratically). Exponential fits should be done and the quality of the fits should be analyzed if the authors want to assert that the declines are exponential. Related to this, the assertion that there is "no offset" in panel d is unsupported. A linear fit should be done, and analysis of the residuals would inform whether there is an offset.

5. In Section 3.3, note that much of the discussion is also relevant to cohort-based models (or at least the ED family). The ED approach involves discretization of a partial differential equation (equation 5 in Moorcroft et al. 2001), and thus there are again questions of the optimal number of age bins, whether the bins should be of different or equal sizes, and criteria for merging.

Technical corrections

P1, L8: do you mean "simulation" rather than "implementation"? This paper, of course, deals with the simulations rather than actual implementations of forest management.

P1, L9: not clear what "hierarchy" is being referred to here

P2, L11: replace "extend" with "extent"

P2, L13-16: there are a couple of sentences where a plural verb "are" is used with a singular subject ("one example")

P3, L5: this sentence seems to have missing words or typos

P3, L6: I am comfortable with the idea that this is a frequently applied approach, but do you have evidence that this is the "most frequently" applied approach?

P3, L30: Perhaps instead of "In this paper we try to", use "The objective of this paper is to"

P7, L4: Note that "data" is plural. Hence, "these data".

Throughout: My sense is that the word "exemplary" is not being used appropriately in the text. Exemplary denotes a particularly good example, whereas I think the authors are oftentimes just referring to an example of the typical sort.

References

Moorcroft, P.R., Hurtt, G.C. and Pacala, S.W., 2001. A method for scaling vegetation dynamics: the ecosystem demography model (ED). Ecological monographs, 71(4), pp.557-586.

Santanello Jr, J.A., Dirmeyer, P.A., Ferguson, C.R., Findell, K.L., Tawfik, A.B., Berg, A., Ek, M., Gentine, P., Guillod, B.P., van Heerwaarden, C. and Roundy, J., 2018. Land–atmosphere interactions: the LoCo perspective. Bulletin of the American Meteorological Society, 99(6), pp.1253-1272.

---

## Referee Comment (RC2) · Anonymous Referee #2 · 13 May 2019

Nabel and colleagues presented in this paper a new development made in the land surface model JSBACH that accounts for sub-grid even-aged forest age classes or cohorts that are outcomes of stand-replacing events such as clear-cut or stand-replacing forest fires. Then they performed a simulation from 1860 from scratch, driven by clearcut-like forest harvest data to match a forest age distribution map recently constructed. They compared simulated results with observations for GPP, LAI and aboveground biomass and model results with and without age classes. As shown in their Fig S2 to S4, model simulation with age class exclusively reduced the magnitudes of all three variables considered, which is expected as the original model structure without age class maintains all these variables close to their equilibrium level (quasi old-growth) if the disturbed area is not big enough to yield a clear decrease. Consequently, for regions where original model shows a high bias, the new development brings the simulation closer to the observation.

I believe the model development direction represented in this paper is worth publication. But the evidences and results presented in it are more like an internal technical report rather than a scientific publication (see my first major comment and other minor comments below). Several aspects of the paper have to be improved if it is to be accepted, which I list as below:

(1) The description of model "development" part needs to be strengthened. In general, the authors failed to highlight their technical advances or difficulties in contrast to the original model version (or the complexity the development that's achieved) to allow reviewer to appreciate their work. I agree this hierarchical structure in JSBACH is novel and it seems to facilitate the management of which processes to be executed on which level to save computation time (as the authors claimed but is not actually shown). Is this feature out of the development by the authors in this work? Otherwise it seems that the authors just made use of this existing model feature and did some simple configuration changes (mainly the number of age classes and their distribution over age) and they claimed this as a new "development". For "development", I would understand as substantive new features in the model, or improvement in parameterization, or new method for model calibration etc. It seems the only paragraph that's fully dedicated to the "development", or the description of the author's new work is the first paragraph in Section 2.3. All other material in the "Methods" section is devoted to introducing JSBACH model structure (2.1), or existing hierarchical model feature (2.2) or simulation set-up (2.4). With this it's hard to appreciate what's really achieved in this paper in terms of "model development". The whole paper more sounds like testing a configuration of the model in terms of age class and performing some sanity check in GPP, LAI and AGB. That's how I reach the feeling of an interval technical report.

(2) There is great confusion in this hierarchical model structure and the advantages that the authors claimed to have. If this overall, sharing model "overhead" can really save computation time, we would expect a non-linear relationship in Fig. 6d ? A decreasing amount of extra time used for each unit increase in number of age class should be expected. From this, I don't see the author's claim that such a feature that different age classes share some common "overhead" process to be computationally efficient as being proved.

(3) Relate to the above point. The authors mentioned throughout the paper the importance of biophysical feedbacks of forest management but nothing of this aspect is shown in the paper. Instead, there is little description on how such processes are simulated in the age-class model structure. The only text I found that gives such similar description seems to be lines 10-13 in Section 2.2 but this is quite vague. The readers are left wandering in what processes belong to "overhead" and which are age-class specific ? For example, how the processes like albedo, energy balance, soil water processes, carbon allocation are simulated ? Which of these processes are "overhead" and how flexible they are in terms of being simulated on different levels ? These are critical for the age-class feature to really reflect forest management impacts but are unfortunately little described.

(4) The essence of the new age-class feature is to yield lower estimate of LAI, GPP and AGB than the old version. Comparing the overall agreement between the old and new feature with observation is nice but not the most convincing way from my point of view, because the old version can always be adjusted/parameterized to agree with the observation and if this is the case, the new version would show a prevalent low bias. What would be nice is to show whether the model improvement is systematically related to the forest age. For example, is the bias or error reduction more pronounced in regions where young forests dominate ? What the processes driving such a decrease in simulated LAI, GPP and AGB and how does this relates to the "ageing" process in the model ? The author mentioned several times of this "ageing" process but what is it and how does it impact the simulation of these variables ? Are examples of new model behaviour related to age-class development is necessary to understand this ? Another way to show the influence of this new development is to show its impact on estimated global fluxes, such as land use change emissions as the authors described in the introduction.

**Some minor and editorial comments:**

- P 3 line 5 : "to extent" -> extend
- P4 line 12: "be able to" could be removed.
- P4 line 11: "a dependency of the maximum leaf area index (LAI) on the available leaf carbon ", what do you mean by "available leaf carbon", does it mean existing leaf biomass or NPP that's allocated to leaf ? I would think it is rather natural and reasonable that maximum LAI being limited by leaf biomass ? How do this feature relate to the age class development ? Is this feature already satisfying for age class structure, or not ?
- P6 line 2: is the "git" feature relevant here, it has been mentioned several times including the in the title.
- P 6 line 4: the upper-bound of what ?
- P 6 line 20: "initiated" can be removed.
- P 6 line 21-22: "which are directed and scheduled on the PFT level but exerted on the ACs ". I don't get the meaning, could it be explained in an easier way ?
- P 7 line 4: Some brief introduction on GPP and LAI data is needed. A critical issue here: as far as I understand Tramontana et al. 2016 GPP data does not consider forest age and it's questionable to use this as a product to evaluate a model with age effect because the age is the key point here.  A recent paper by Besnard et al. ERL

([https://doi.org/10.1088/1748-9326/aaeaeb](https://doi.org/10.1088/1748-9326/aaeaeb)) tried to address this but I don't know whether they have GPP product. Likewise, is the LAI data pure satellite observation ?
- P 8 line 24 : "to be harvested fraction" -> to-be-harvested-fraction ? A noun form should be here but please check.
- Figure 2: what's the "UML" ?
- Figure 3: $AC_M$ , I would use $AC_i$, which distinguishes clearly with $AC_N$,i.e., the former refers to a common AC, while the latter refer to the old-growth AC.
- Figure 5: Label for vertical axis not consistent with others. Can you use more expressive label, for example, "Normalized RMSE?".
- Equation (2): I would write simply N-1 for the denominator…
- P12 line 1: "as also discussed" -> as is also discussed
- Figure 7, caption: "Stars mark the JJA GPP per age-class", please indicate this is for simulated data. Could you somehow simply the caption ? It's rather long and almost deters reading.

---

## Author Comment (AC1) · 22 Nov 2019

Thank you for your review and your comments that help to clarify the manuscript. Below we duplicate your comments (**bold**) and respond to them point-by-point (*italics*) followed by modifications that will be adopted in the revised manuscript.

1. The text raises two important expectations that were not fulfilled in the results. First is the issue of land-atmosphere interactions. I would argue that it is conventional to think of land-atmosphere interactions in terms of energy and water budgets, and indeed some authors (e.g., Santanello Jr et al. 2018) define land-atmosphere interactions exclusively in terms of energy and water budgets. Yet the current manuscript contains no results related to energy and water budgets. Personally, I think that such results would be an interesting addition to the paper. Without such an addition, I think the authors need to change the text to more specifically refer to carbon fluxes and budgets.

Thank you for pointing this out. We adapted the text in several places to not raise the expectation of presenting results related to energy and water budgets (just for variables influencing energy and water budgets, in particular LAI). We further tested the response of evapotranspiration (ET) and we attached a figure showing the change in NRMSE for ET to the review responses (Fig. R1). Figure R1 shows that the shape of the change in NRMSE is comparable to that for LAI, however, the magnitude is smaller. We thus feel that the ET plot would not add much information and decided not to show the figure in the manuscript.

Examples of sentences changed in the text: The second and third sentence of the abstract now state

Forest age-structures in turn influence biophysical and biogeochemical interactions of the vegetation with the atmosphere key land surface processes, such as photosynthesis and thus the carbon cycle. Yet, many dynamic global vegetation models (DGVMs), including those used as land surface models (LSMs) in Earth system models (ESMs), do not account for subgrid forest age structures, despite being used to investigate land-use effects on the global carbon budget or simulating land-atmosphere interactions-biogeochemical responses to climate change.

The second sentence of the summary and outlook section now states:

JSBACH4–FF allows land-atmosphere interactions key land surface processes to be simulated in dependence of forest age and, simultaneously, to trace the exact forest age, enabling the which is a precondition for any implementation of age-based forest management schemes in JSBACH4–FF.

Second is the issue of forest management. The authors repeatedly state (see P11,L10-11 for one example) that an advantage of the model is in simulating different forest management scenarios. However, this is not exploited in the paper (I know that the forest management scenarios in Fig. 7 are different, but so is the climate, so one cannot isolate the effect of the management scenario). Why not illustrate the power of the new modelling approach by running different forest management scenarios for a single grid cell?

The main purpose of the paper was the description of the implementation of forest age-classes in JSBACH4 and the presentation of the applied new approach for this introduction of forest age-classes. The possibility to implement different forest management scenarios is one important motivation for this model development, but neither the only motivation, nor a focus of our paper and we consider running the model for several more forest management scenarios beyond the scope of this study. In order to not suggest the focus of our study is studying effects of various forest management scenarios we adapted the text in several places. However, we would like to note that technically we in fact do compare two different forest management scenarios, namely the average harvest in the PFT simulation and the harvest of the oldest forest area in the age-class simulations (under the same climate).

**Examples for adapted text passages - line 8-9 first page:**

Our scheme combines The first being a computationally efficient age-dependent simulation of all relevant processes, such as photosynthesis and respiration, without loosing the information about using a restricted number of age-classes. The second being the tracking of the exact forest age, which is a prerequisite for the any implementation of age-based forest management.

**First sentence of the results and discussions section:**

Having forest age-classes in JSBACH4–FF facilitates a finer discretisation in each grid-cell and enables the is a precondition for any implementation of age-based forest management.

**Additionally we inserted a statement on the two applied forest management schemes in the Section on harvest maps (now 2.3.4):**

In different simulation types – with or without age-classes – the same harvest maps were used, but different forest management schemes were applied. In simulations with age-classes, a clear-cut according to the fractions in the harvest map was taken from the oldest age-class. In the simulation without age-classes, the PFT simulation, we used the same harvest fractions as in the simulations with age-classes, but harvest was applied as done in JSBACH3 (Reick et al., 2013), i.e. by diluting the wood carbon of the harvested PFT tile.

2. Some of the methods were not adequately justified. First, I am concerned that the initial condition is unrealistic. Why did simulations begin in 1860 from bare ground rather than from a spin-up? Of course, the 1860 initial condition would more realistically be represented by many forested areas.

Thanks for pointing this out, we have now added our reasoning to start from scratch in 1860:

Simulations started in 1860 from scratch, i.e. with empty vegetation carbon stocks, and were run up to 2010. Empty carbon stocks are a simplification used in the absence of global knowledge on the state of the forest in 1860, but have no influence on our results, since in simulations with JSBACH4 (4.20p7) LAI, GPP and AGB only depend on the age since the last clearing event, not on the history before that. The starting date of 1860 was chosen such that it covers at least one full cycle of regrowth, as the oldest age resolved in the simulations matches that of the observation-based data (Poulter et al., 2018).

Note that the used JSBACH version has no influence of soil carbon and nutrient state on vegetation growth, which could be influenced by the history prior to the last clearing event,.

**Second, I could not determine from the paper how one goes from the 2010 age-class distribution to time-dependent (1860-2010) harvest rates. This procedure should be described.**

We did outline the derivation of the annual harvest maps in 2.4.3. (now 2.3.4), and described the procedure in detail in the supplementary material S2.For the procedure deriving the harvest rates we politely point the referee to the listing in S2.

Third, it seems like there is an inconsistency between the definition of the model's PFTs (tropical evergreen and deciduous, extratropical evergreen and deciduous) and the Poulter et al. PFTs (broadleaf evergreen and deciduous, needleleaf evergreen and deciduous). What is the correspondence?

We added the mapping formula to 2.4.3 were we previously only stated that the Poulter et al. (2018) PFTs were mapped to JSBACH's PFT cover fractions.

**Part of the edited text in 2.4.3:**

The map by Poulter et al. (2018) provides a grid with 0.5 ° resolution of the global forest age distribution of 4 forest PFTs(needleleaf evergreen and deciduous four forest PFTs: needleleaf evergreen (NE) and needleleaf deciduous (DE), as well as broadleaf evergreen and deciduous) on a grid with 0.5resolution (BE) and broadleaf deciduous (BD). The map uses a discretisation into 15 age-classes, covering 10 years each, with the last class containing all area with an age >140 years. In a pre-processing step, the map was remapped to T63 using the conservative remapping operator of the CDOs. Subsequently, the PFTs from the map were scaled to area sums of the two evergreen and the area sums of the two deciduous PFTs from Poulter et al. (2018) were used to derive the age-class maps for JSBACH's PFT cover fractions. From these scaled evergreen and deciduous PFTs, respectively, following Eq. 6

3. The comparison to observations can be made more substantial. RMSE is a helpful statistic, but I wonder what is being missed by only considering this statistic. For example, I wonder what can be learned from Taylor diagrams? I am certainly not asking that the paper include Taylor diagrams for every variable, but rather such diagrams could be analyzed in a preliminary analysis and the most exciting ones presented in the paper or supporting information.

Following this suggestion we performed an analysis using Taylor diagrams comparing the PFT simulation, *i.e.* the simulation without age-classes with the IAS11 simulation, *i.e.* the age-class set up used in the more detailed comparison in the manuscript. We included the Taylor diagrams in the supplementary and added a note in the methods section. However, since we found that the Taylor diagrams do not allow new, important conclusions to be drawn beyond the NRMSE comparison, we did not include further analysis of the Taylor plots in the results section.

**Text added to Section 2.2 (previously 2.2)**

**In addition, we created Taylor diagrams for each variable, season and region (see Figures S5.5–S5.11 in the supplementary).**

4. There are some problems with the interpretation of the results. First, I think it misses the point to repeatedly state that the new model is better. Rather, the fundamental result is that new model tends to reduce GPP and LAI relative to the old model. The new model is better because the old model was biased high. If the old model had been unbiased, then the new model would have been biased low. Alternatively, suppose that there is another modeling group excited by this study, and that that modeling group has a model that is biased low. Then implementation of this scheme would probably make that model worse.

We agree that our implementation of age-classes in a model with a low bias would probably make such a model even worse and adapted the manuscript to stress that introducing the age-classes in a model which is biased low would lead to an increase in the comparison error. We also agree that results closer to observations could just be a consequence of compensating for a high bias (for whatever reason that high bias may have existed) in the old model and now distinguish between model improvement in terms of quantitative results (which could be disputed) and model improvement in terms of inclusion of processes known to exist in reality (in the latter respect the new model is clearly "better"). Nevertheless, we would like to point out that spatially explicit comparisons of the results from the PFT simulation and observation-based data ("OBS-PFT" in Figs. S4.2–S4.4, column 2) indicate several areas of underestimation (red) and of overestimation (blue) for all variables, thus the old model was not merely biased high.

In addition, we have tried to understand if the high bias in the old model is due to not including ageclasses or due to other processes. We found that indeed part of the high bias stems from missing an adequate representation of regrowth: we attached a figure showing the change in model bias per mean age to the review responses (Fig.R2). Figure R2 shows that the change in model bias decreases with forest age, indicating that the error reduction happens where the old model was biased high due to not considering forest age.

In the results and discussions section looking at the benefit of having age-classes (3.2) we inserted the following summary and caveat:

In summary, simulations using age-classes led to a decrease in the simulated GPP, LAI and AGB values due to their non-linear increase with a saturation for older ages. This caused a decrease in the NRMSEMax-Min in areas where the PFT simulation was biased high and an increase in the NRMSEMax-Min in areas where the PFT simulation was biased low. Thus, if such a forest age-structure would be implemented in a DGVM being predominately biased low, the difference to the observation-based data could increase.

Second, I think that more care needs to be taken in the interpretation of Figure 6. While the curves in panels a-c are decreasing, the authors do not quantitatively support their assertion that the curves are decreasing exponentially (and not, say, quadratically). Exponential fits should be done and the quality of the fits should be analyzed if the authors want to assert that the declines are exponential. Related to this, the assertion that there is "no offset" in panel d is unsupported. A linear fit should be done, and analysis of the residuals would inform whether there is an offset.

Concerning the shape of the curves in panels a-c: We eliminate statements about the shape of the curve, since this is not relevant for our conclusions. The second last sentence of the abstract, for example, now states:

The comparisons show differences exponentially decreasing with the decreasing differences and increasing computation costs with an increasing number of distinguished age-classes and linearly increasing computation costs.

Concerning computing time (panel d): The pre-last paragraph of the Evaluation section (3.1) now reads:

Comparisons of required CPU times show a linear near-linear increase with an increased number of ageclasses (Fig. 6d) and neither a difference between the two age distribution schemes, nor an striking offset as compared to the PFT simulation. This behaviourA near-linear increase with an increased number of age-classes was expected, since the processes requiring most of the computing time, such as the calculation of photosynthesis, carbon allocation and respiration, are conducted on the age-classes. The absence of an striking offset comparing the PFT simulation with the age-class simulations indicates that the introduced organisational overhead on the PFT level in simulations with age-classes is not substantial, i.e. tracing of the exact forest age and redistributions of area fractions and other state variables among tiles, is not dominating the computation times.

5. In Section 3.3, note that much of the discussion is also relevant to cohort-based models (or at least the ED family). The ED approach involves discretization of a partial differential equation (equation 5 in Moorcroft et al. 2001), and thus there are again questions of the optimal number of age bins, whether the bins should be of different or equal sizes, and criteria for merging.

Thank you for this comment.

**Technical corrections**

P1, L8: do you mean "simulation" rather than "implementation"? This paper, of course, deals with the simulations rather than actual implementations of forest management.

We actually meant implementation. We split and rephrased the sentence, now stating:

In this paper we present a new scheme to introduce forest age-classes in hierarchical tile-based DGVMs combining benefits of recently applied approaches. Our scheme combines The first being a computationally efficient age-dependent simulation of all relevant processes, such as photosynthesis and respiration, without loosing the information about using a restricted number of age-classes. The second being the tracking of the exact forest age, which is a prerequisite for the any implementation of age-based forest management.

**P1, L9: not clear what "hierarchy" is being referred to here**

Thank you. We edited this sentence.

This combination is achieved by using the tile-hierarchy to track the area fraction for each age on an aggregated plant functional type level, whilst simulating the relevant processes for a set of age-classes.

**P2, L11: replace "extend" with "extent"**

Changed accordingly.

P2, L13-16: there are a couple of sentences where a plural verb "are" is used with a singular subject ("one example")

Changed accordingly.

**P3, L5: this sentence seems to have missing words or typos**

Thank you, we corrected "extent" to "expand".

**P3, L6: I am comfortable with the idea that this is a frequently applied approach, but do you have evidence that this is the "most frequently" applied approach?**

We edited the sentence and now state that it is the more commonly used (of the two recently developed approaches that we present, based on the number of references that we found and list using one or the other approach):

To extent expand tile-based DGVMs to represent subgrid forest age structures, two approaches have recently been developed. The most more frequently applied approach has been to increase the number of tiles in such a way that a certain number of age-classes or structurally similar stands can be distinguished.

**P3, L30: Perhaps instead of "In this paper we try to", use "The objective of this paper is to"**

We edited the sentence, it now states:

In this paper we try tobridge the two approaches for extending tile-based DGVMs to represent subgrid forest age in the sense that we present a way to trace the actual age of the forests in a grid-cell despite following the first approach using a restricted number of additional tiles and thus required merges.

**P7, L4: Note that "data" is plural. Hence, "these data".**

Changed to "datasets".

Throughout: My sense is that the word "exemplary" is not being used appropriately in the text. Exemplary denotes a particularly good example, whereas I think the authors are oftentimes just referring to an example of the typical sort.

We replaced the three occurrences of exemplary in the manuscript.

Figure R1: Change in NRMSEMax-Min when comparing simulated evapotranspiration (ET) of simulations using an increasing number of age-classes to observation based ET data (GLEAM V2A - Miralles et al., 2011). As in the comparison for GPP, LAI and AGB (Supplementary figure S3.1) averaging has been conducted giving equal weights to each of the four seasons.

Figure R2: Change in the difference of the differences of simulated and observed MAM LAI using the PFT and the IAS11 simulation. Dots mark differences in single grid-points at grid-point mean ages, the red line marks the mean change of all dots for each mean-age (rounded to integer years). Note the difference in the y-axis of the two panels.

**References**

Miralles, D. G., et al. "Magnitude and variability of land evaporation and its components at the global scale." (2011).

Change in MAM LAI bias (abs(OBS-PFT)-abs(OBS-IAS11))

---

## Author Comment (AC2) · 22 Nov 2019

Thank you for your review and the suggestions that help to clarify the manuscript. We have duplicated your comments (**bold**) below, each followed by a point-by-point response (*italics*) including the modifications that will be adopted in the revised manuscript.

**(1)The description of model "development" part needs to be strengthened. In general, the authors failed to highlight their technical advances or difficulties in contrast to the original model version (or the complexity the development that's achieved) to allow reviewer to appreciate their work. I agree this hierarchical structure in JSBACH is novel and it seems to facilitate the management of which processes to be executed on which level to save computation time (as the authors claimed but is not actually shown). Is this feature out of the development by the authors in this work? Otherwise it seems that the authors just made use of this existing model feature and did some simple configuration changes (mainly the number of age classes and their distribution over age) and they claimed this as a new "development". For "development", I would understand as substantive new features in the model, or improvement in parameterization, or new method for model calibration etc. It seems the only paragraph that's fully dedicated to the "development", or the description of the author's new work is the first paragraph in Section 2.3. All other material in the "Methods" section is devoted to introducing JSBACH model structure (2.1), or existing hierarchical model feature (2.2) or simulation set-up (2.4). With this it's hard to appreciate what's really achieved in this paper in terms of "model development". The whole paper more sounds like testing a configuration of the model in terms of age class and performing some sanity check in GPP, LAI and AGB. That's how I reach the feeling of an interval technical report.**

*Thank you for this important comment. We rewrote large parts of Section 2.1 and the former Section 2.3 (now 2.2) and adapted Fig.2 and Fig.3 to better explain the newly developed scheme and to better emphasise new model developments. The tile hierarchy did indeed already exist, however, as a purely infrastructural piece of code. For each of the introduced process this infrastructure had to be extended. In particular, we newly introduced the age vector to track the age, and the processes managing the age-classes. These processes were either newly implemented (ageing and harvest) or had to be advanced (disturbances). An additional major technical advancement was to address the new necessity to introduce shifts of area fractions from one AC to another, as well as resulting shifts of forest carbon and the re-determination of other affected state variables. We now also describe these new infrastructural developments.*

**(2)There is great confusion in this hierarchical model structure and the advantages that the authors claimed to have. If this overall, sharing model "overhead" can really save computation time, we would expect a non-linear relationship in Fig. 6d ? A decreasing amount of extra time used for each unit increase in number of age class should be expected. From this, I don't see the author's claim that such a feature that different age classes share some common "overhead" process to be computationally efficient as being proved.**

*Thank you for pointing this out. We assume that different concepts got mixed up, being (1) what we called the "organisational overhead", (2) savings of computation time by only introducing a restricted number of age classes, and (3) the potential to save computation time by simulating processes on different levels of the tile-hierarchy. When rewriting Section 2.1 and the former Section 2.3 (now 2.2), as well as upon adding to Fig.2 and Fig.3, we attempted to resolve this confusion. We particularly removed statements targeting point (3) listed above since these are not directly connected to the developments presented in our paper. For clarification: we used "organisational overhead" to refer to the additional computation time required for bookkeeping of the exact forest age and managing merges of fractions of different age-classes.*

*In addition to rewriting Sections 2.1-2.3 (now 2.1 and 2.2) we explain what we term the "organisational overhead" on its first occurrence in Section 3.1, stating:*

The absence of a striking offset comparing the PFT simulation with the age-class simulations indicates that the introduced organisational overhead on the PFT level in simulations with age-classes , i.e. tracing of the exact forest age and redistributions of area fractions and other state variables among tiles, is not dominating the computation times.

**(3)Relate to the above point. The authors mentioned throughout the paper the importance of biophysical feedbacks of forest management but nothing of this aspect is shown in the paper. Instead, there is little description on how such processes are simulated in the age-class model structure. The only text I found that gives such similar description seems to be lines 10-13 in Section 2.2 but this is quite vague. The readers are left wandering in what processes belong to "overhead" and which are age-class specific ? For example, how the processes like albedo, energy balance, soil water processes, carbon allocation are simulated ?**

*When rewriting Section 2.1 and the former Section 2.3 (now 2.2) we attempted to also resolve this point changing the text to make more explicit (1) which processes already were present in the basis version of JSBACH4 and which have been implemented in the course of this study and (2) on which level of the hierarchy the different processes are executed. Regarding raised expectations concerning biogeophysical feedbacks (which was also commented by reviewer 1, comment 1) we adapted the text in several places to not raise the expectation of presenting results related to energy and water budgets (just for variables influencing energy and water budgets, in particular LAI). We further tested the response of evapotranspiration (ET) and we attached a figure showing the change in NRMSE for ET to the responses to reviewer 1 (Fig. R1). Figure R1 shows that the shape of the change in NRMSE is comparable to that for LAI, however, the magnitude is smaller. We thus feel that the ET plot would not add much information and decided not to show the figure in the manuscript.*

**Which of these processes are "overhead" and how flexible they are in terms of being simulated on different levels ? These are critical for the age-class feature to really reflect forest management impacts but are unfortunately little described.**

*We made several text changes, particularly in the former Section 2.3 (now 2.2), as well as additions to Fig. 2 and Fig. 3, to make more explicit which processes and state variables are located on which level (please also see the response to point 1). Regarding the term "overhead", please also refer to response 2.*

**(4)The essence of the new age-class feature is to yield lower estimate of LAI, GPP and AGB than the old version. Comparing the overall agreement between the old and new feature with observation is nice but not the most convincing way from my point of view, because the old version can always be adjusted/parameterized to agree with the observation and if this is the case, the new version would show a prevalent low bias.**

*We agree that the new model version does show a better performance where the old model version was biased high, particularly in several of the regions having young forests. Nevertheless, we would like to point out that the improved model performance is related to not considering forest age (see Fig.R2 in the responses to reviewer 1). Furthermore, we would like to point out that spatially explicit comparisons of the results from the PFT simulation and observation-based data ("OBS-PFT" in Figs. S4.2–S4.4, column 2) indicate several areas of underestimation (red) and of overestimation (blue) for all variables, thus the old model was not merely biased high. Tuning the old version could potentially result in a lower high bias in regions where the forest is young, but this will result in a low bias in regions where the same PFT is mature. So, tuning is not an alternative to including age classes. In order to raise awareness of the general direction of biases (which was also commented on by reviewer 1, comment 4) we inserted the*

*following summary and caveat in the results and discussions section looking at the benefit of having age-classes (3.2):*

In summary, simulations using age-classes led to a decrease in the simulated GPP, LAI and AGB values due to their non-linear increase with a saturation for older ages. This caused a decrease in the $NRMSE_{Max-Min}$ in areas where the PFT simulation was biased high and an increase in the $NRMSE_{Max-Min}$ in areas where the PFT simulation was biased low. Thus, if such a forest age-structure would be implemented in a DGVM being predominately biased low, the difference to the observation-based data could increase.

**What would be nice is to show whether the model improvement is systematically related to the forest age. For example, is the bias or error reduction more pronounced in regions where young forests dominate? What the processes driving such a decrease in simulated LAI, GPP and AGB and how does this relates to the "ageing" process in the model? The author mentioned several times of this "ageing" process but what is it and how does it impact the simulation of these variables? Are examples of new model behaviour related to age-class development is necessary to understand this? Another way to show the influence of this new development is to show its impact on estimated global fluxes, such as land use change emissions as the authors described in the introduction.**

*For the relation of the improvement in model performance and forest age please refer to Fig. R2 in the responses to reviewer 1. Regarding the simulation of LAI, GPP and AGB: their relation to the "ageing process" stems from LAI, GPP and AGB being simulated separately for each age-class. Due to the non-linear relationship of GPP, LAI and AGB with forest age (Fig.7 of the manuscript) simulations of a mixed aged forest will result in higher values in a mean age forest simulation (PFT) than in a simulation resolving different age-classes and thus leading to independent simulations of LAI, GPP and AGB on tiles representing different forest ages. Regarding the explanation of the ageing process: we added a more detailed description of the process in 2.2:*

**Ageing** The newly implemented process of forest "ageing" happens annually: upon ageing each tracked forest fraction gets one year older. Yet, a shift from one age-class to the next age-class only happens for the area of the oldest age ($maxA_{K-1}$-1) of an age-class $AC_{K-1}$, i.e. only the forest area which upon getting one year older exceeds the upper age bound $maxA_{K-1}$ of the $AC_{K-1}$ needs to be shifted into $AC_K$. Thanks to the tracking of the age in the *fractPerAge* vector, the exact area fraction with age $maxA_{K-1}$-1 is known.

**Some minor and editorial comments:**

**P 3 line 5 : "to extent" -> extend**

*Changed accordingly.*

**P4 line 12: "be able to" could be removed.**

*The sentence has been adapted (see response to the next comment).*

**P4 line 11: "a dependency of the maximum leaf area index (LAI) on the available leaf carbon ", what do you mean by "available leaf carbon", does it mean existing leaf biomass or NPP that's allocated to leaf? I would think it is rather natural and reasonable that maximum LAI being limited by leaf biomass? How do this feature relate to the age class development ? Is this feature already satisfying for age class structure, or not ?**

*We agree with the referee that it is natural and reasonable that the maximum LAI, i.e. the LAI that can maximally be reached at the peak of a season, is limited by leaf biomass. However, this is not the case in the standard JSBACH3 (Mauritsen et al., 2019) version and therefore also not in the standard JSBACH4 version. In JSBACH3, the maximum LAI is a PFT-dependent constant, which is why we implemented this dependency in an independent study (Naudts et al., in prep.). We now explicitly stress that this was not the case in JSBACH3 and that it is a precondition for the introduction of our age-classes. We rewrote this part of Section 2.1 now stating:*

As an important amendment to the current version (4.20p7) , we ported a new JSBACH3 development, which we implemented in a recent independent study (Naudts et al., in prep.): While previous JSBACH3 versions assumed a PFT-dependent but constant maximum leaf area index (LAI), that is the LAI value that can maximally be reached at the peak of a season, Naudts et al. (in prep.) introduced a dependency of the maximum LAI on the available leaf biomass. Such a dependency is a prerequisite for simulating forest re-growth and thus for the introduction of age-classes.

**P6 line 2: is the "git" feature relevant here, it has been mentioned several times including the in the title.**

*We prefer to keep this information for reproducibility reasons.*

**P 6 line 4: the upper-bound of what ?**

*Thank you for pointing this out. We edited this (and other) sentences. This sentence now states:*

In addition, a to be pre-defined upper age bound per age-class $AC_M$ $_K$ $(maxA_K)$ as well as a total maximum age (maxAge) were introduced.

**P 6 line 20: "initiated" can be removed.**

*This sentence changed upon rewriting of Section 2.3 (now 2.2).*

**P 6 line 21-22: "which are directed and scheduled on the PFT level but exerted on the ACs ". I don't get the meaning, could it be explained in an easier way ?**

*This sentence changed upon rewriting of Section 2.3 (now 2.2).*

**P 7 line 4: Some brief introduction on GPP and LAI data is needed. A critical issue here: as far as I understand Tramontana et al. 2016 GPP data does not consider forest age and it's questionable to use this as a product to evaluate a model with age effect because the age is the key point here. A recent paper by Besnard et al. ERL (https://doi.org/10.1088/1748-9326/aaeaeb) tried to address this but I don't know whether they have GPP product. Likewise, is the LAI data pure satellite observation?**

*Thank you for pointing this out. We now briefly introduce GPP and LAI in Section 2.3.1. Furthermore, we added some caveats regarding the observation-based data in Section 3.2. Additionally, we redid Fig.7 in the text using LAI instead of GPP.*

*Brief introduction in Section 2.2.1 (former 2.3.1):*

We used 2010 MODIS LAI (Myneni et al., 2002) and GPP data obtained from machine learning methods trained on flux-tower measurements (Tramontana et al., 2016).

*Added sentences in Section 3.2:*

In this context, caveats regarding the observation-based data themselves need to be raised. A known caveat regarding MODIS LAI data is the problem of reflectance saturation in dense canopies making the reflectance insensitive to changes in LAI (Myneni et al., 2002). This problem, which is particularly relevant to the tropical region, could lead to a general high bias of the model compared to the observation-based data. However, since this problem is more typical for denser old grown forests, this high bias would also occur in the simulations with age-classes. Regarding the GPP data from Tramontana et al. (2016), a recent study by Besnard et al. (2018) criticised that the applied empirical upscaling techniques do not directly consider forest age, making it unclear how well they can capture age-related dynamics. In their study, Besnard et al. (2018) advocate the development of alternative global datasets considering forest age as a predictor.

**P 8 line 24 : "to be harvested fraction" -> to-be-harvested-fraction ? A noun form should be here but please check.**

*The sentence was superfluous and has been deleted.*

**Figure 2: what's the "UML" ?**

*Now spelled out (Unified Modeling Language).*

**Figure 3: AC M , I would use AC i , which distinguishes clearly with AC N ,i.e., the former refers to a common AC, while the latter refer to the old-growth AC.**

*Thank you very much for this suggestion. We updated the figure accordingly (for better readability we used K instead of I). We also replaced all occurrences of AC M in the text by AC K.*

**Figure 5: Label for vertical axis not consistent with others. Can you use more expressive label, for example, "Normalized RMSE?".**

*Fig.5 shows the NRMSE_{Max-Min} for each variable, region and season. Fig.6 and Fig. S3.1 show averages over the seasons or the seasons and the regions, respectively. Therefore, the y-axis of these figures are labeled differently. We added "normalised root mean squared error" to the figure captions.*

**Equation (2): I would write simply N-1 for the denominator...**

*In the former EQ.2 (now EQ.4) the denominator should contain the sum of i's (i.e. for N=5: 1+2+3+4 = 10). However, there had been a mistake in the equation (the sum over the i's started with i), which we now corrected.*

**P12 line 1: "as also discussed" -> as is also discussed**

*Changed accordingly.*

**Figure 7, caption: "Stars mark the JJA GPP per age-class", please indicate this is for simulated data.**

*Changed accordingly.*

**Could you somehow simply the caption ? It's rather long and almost deters reading.**

*We tried to shorten the caption, but it still remains long as we prefer to include all the information necessary to read the figure. The caption now reads:*

Example grid-points comparing 2001-2010 mean spring leaf area index (MAM LAI) from simulation without (PFT) and with age-classes (IAS11) to observation-based data. The map in the center shows the difference of differences between the observation-based data and the simulations (abs(OBS-PFT)-abs(OBS-IAS11)), i.e. it shows where the results from the simulation with age-classes (IAS11) deviate less (blue) or more (red) from the observation-based data than the PFT simulation results (see also Figs. S4.2-S4.4, column 4). Dashed lines in the map mark the three selected regions (see Table 2). The plots (a-g) show the LAI of selected PFTs (ETD: extratropical deciduous; ETE: extratropical evergreen; TD: tropical deciduous; TE: tropical evergreen) as well as their according area fractions per age-class and per year at the labelled grid-points. Center latitude, longitude and grid-cell cover fraction (cf) of the depicted PFT are indicated. The x-axis reflects the age from 0-151 (purple) with the age-classes (black) indicated at the age centres. The two right y-axes represent the bars: depict are the 2010 area fractions relative to the area of the depicted PFT. Blue bars are per age-class (black y-axes) and depict the fraction of each age-class (i.e. one bar per age-class); the yellow framed purple bars depict the fraction of each age (i.e. one bar per year). The left y-axis depicts the LAI. Stars mark the simulated LAI per age-class, and the lines the LAI of the depicted PFT – blue dashed line: IAS11 simulation, black line: PFT simulation, green line: 2010 value from the observation-based data. Note: 1. The age-class LAI is only depicted for age-classes having non-zero fractional cover over the whole timespan 2001-2010 (this is not the case for the age-classes 9 and 10 in panel c, f and g). 2. Age and age-class fractions of classes 2-8 in panel g are very small and therefore not visible above the x-axis. 3. Since we did not apply any harvest in the final simulation year 2010, the first year and accordingly the youngest age-class are always empty.